# Experiences with sexual orientation and gender identity conversion therapy practices among sexual minority men in Canada, 2019–2020

Travis Salway[1,2,3,4]*, Stephen Juwono[1,4,5], Ben Klassen[4], Olivier Ferlatte[4,6,7], Aidan Ablona[2,4], Harlan Pruden[2,4], Jeffrey Morgan[4,8], Michael Kwag[4], Kiffer Card[4,9], Rod Knight[4,10], Nathan J. Lachowsky[4,9]

1 Faculty of Health Sciences, Simon Fraser University, Burnaby, British Columbia, Canada, 2 British Columbia Centre for Disease Control, Vancouver, British Columbia, Canada, 3 Centre for Gender and Sexual Health Equity, Vancouver, British Columbia, Canada, 4 Community-Based Research Centre, Vancouver, British Columbia, Canada, 5 Faculty of Science, University of British Columbia, Vancouver, British Columbia, Canada, 6 l'École de santé publique de l'Université de Montréal, Montréal, Québec, Canada, 7 Centre de Recherche en Santé Publique, Montréal, Québec, Canada, 8 School of Population and Public Health, University of British Columbia, Vancouver, British Columbia, Canada, 9 University of Victoria, Victoria, British Columbia, Canada, 10 Department of Medicine, University of British Columbia, Vancouver, British Columbia, Canada

* travis_salway@sfu.ca

**Data Availability Statement:** Partial Sex Now data is publicly available on the Our Stats dashboard (https://ourstats.ca/our-dashboard). The University

## Abstract

### Background

"Conversion therapy" practices (CTP) are organized and sustained efforts to avoid the adoption of non-heterosexual sexual orientations and/or of gender identities not assigned at birth. Few data are available to inform the contemporary prevalence of CTP. The aim of this study is to quantify the prevalence of CTP among Canadian sexual and gender minority men, including details regarding the setting, age of initiation, and duration of CTP exposure.

### Methods

Sexual and gender minority men, including transmen and non-binary individuals, aged $\geq$ 15, living in Canada were recruited via social media and networking applications and websites, November 2019—February 2020. Participants provided demographic data and detailed information about their experiences with CTP.

### Results

21% of respondents (N = 9,214) indicated that they or any person with authority (e.g., parent, caregiver) ever tried to change their sexual orientation or gender identity, and 10% had experienced CTP. CTP experience was highest among non-binary (20%) and transgender respondents (19%), those aged 15–19 years (13%), immigrants (15%), and racial/ethnic minorities (11–22%, with variability by identity). Among the n = 910 participants who experienced CTP, most experienced CTP in religious/faith-based settings (67%) or licensed healthcare provider offices (20%). 72% of those who experienced CTP first attended before

of Victoria's Human Research Ethics Board has only approved storage of our data on secure university servers since the data contain potentially sensitive information about study participants. Data is available on request through secure university servers only. Any requests to access the data can be made to University of Victoria's Human Research Ethics Board (250-472-4545 or ethics@uvic.ca; reference Ethics Protocol Number BC17-487).

**Funding:** Analyses of secondary data (SexNow) were funded by the Canadian Institutes of Health Research (#168193). The funders had no role in study design, data collection and analysis, decision to publish, or preparation of the manuscript.

**Competing interests:** The authors have declared that no competing interests exist.

the age of 20 years, 24% attended for one year or longer, and 31% attended more than five sessions.

## Interpretation

CTP remains prevalent in Canada and is most prevalent among younger cohorts, transgender people, immigrants, and racial/ethnic minorities. Legislation, policy, and education are needed that target both religious and healthcare settings.

## Introduction

"Conversion therapy" practices (CTP) are organized and sustained efforts to avoid the adoption or expression of lesbian, gay, bisexual, or queer (LGBQ) sexual orientations, gender identities not assigned at birth, and/or non-conforming gender expressions [1, 2]. In the case of CTP targeting sexual orientation, these practices often borrow pseudo-scientific behavioural interventions to supress or deter same-sex/gender attraction (including sex with members of the same sex/gender) [3]. CTP targeting gender identity and expression likewise include so-called psychotherapeutic attempts to "discourage or delay the adoption of gender identities not assigned at birth, as well as non-conforming gender expressions" [4, 5]. Research evidence has clearly established multiple harms associated with CTP, including feelings of shame, anxiety, depression, problematic substance use, suicide ideation, and/or suicide attempts [6–11].

CTP are undergirded by a broader set of practices known as sexual orientation and gender identity or expression change efforts (SOGIECE), which include CTP, as well as less-delineated cissexist and heterosexist stressors experienced by sexual minorities (i.e., those who experience non-heterosexual attraction, behaviours, or identities) and gender minorities (i.e., those who experience gender identities that differ from sex assigned at birth and those who have non-conforming gender expressions) [12]. For further clarity, while CTP tend to have a defined structure—a set number of individual- and/or group-focused sessions—SOGIECE include subtler but often more pervasive attempts to pursuade and affirm rigid expectations of a cis-gender and heterosexual expressions and identities [12]. SOGIECE take a broad variety of forms and are context-specific. For example, in some cases, these change efforts manifest as conversations between a religious leader and believer, in which the leader compels the believer to adhere to heteronormative or cisnormative behaviors. In other cases, institutions and practices that are not typically regarded as CTP nonetheless constitute SOGIECE. For example, residential schools—used by Canadian and US governments and Christian churches to indoctrinate Indigenous children with European ways of living—were also used to enforce European binary notions of gender, by imposing particular styles of dress, hair, and behavior consistent with those expected of boys and girls, thereby erasing Two-Spirit traditions (Two-Spirit is a contemporary umbrella term and organizing tool to facilitate Indigenous Peoples' connections with Nation-specific expressions and roles of gender and sexual diversity) [13, 14].

Data regarding the prevalence and nature of contemporary CTP and SOGIECE are scarce. Recent surveys in the US have estimated that 4–18% of sexual and gender minority people have experienced CTP at some point in their lifetime—with estimates varying based on age of participants, other sample characteristics, and definition of CTP used [10, 15–18]. To date, two similar non-probabilistic venue/network-sampled national surveys have been conducted in Canada, one in 2011–12 focused on sexual minority men (*Sex Now* 2011–12, N = 8388, median age 40–49 years) [8] and one in 2019 focused on gender minorities (*Trans PULSE* 2019, N = 2033, median age 25–34 years) [19]. *Sex Now* 2011–12 defined CTP as "sexual

repair/reorientation counseling" and found that 3.5% of men surveyed had experienced this form of CTP during their lifetime [8]. *Trans PULSE* 2019 defined CTP as "counselling or programs to try to make your gender match with your sex assigned at birth" and found that 11% of those surveyed reported a lifetime experience of CTP [19]. While these data give a sense of the scope of contemporary CTP, numerous gaps in knowledge remain, notably the settings in which CTP occurs, age at initiation of CTP, and duration of CTP. Empirical data regarding these CTP parameters can help to inform and refine legislative, policy, and educational approaches to prevent CTP (including bans and educational campaigns explaining the dangers of CTP), by identifying where and when to target preventive efforts [12, 20].

As international policymakers and legislators are increasingly acknowledging CTP as a threat to the health and well-being of sexual and gender minority people [2, 21], additional, fulsome, and contemporary characterizations of the prevalence and nature of CTP are urgently needed. As of November 18, 2020, four Canadian provinces and territories (Ontario, Nova Scotia, Prince Edward Island, Yukon) and numerous Canadian municipalities (including Vancouver, Calgary, and Edmonton) have enacted a patchwork of legislative CTP bans that range from prohibiting CTP for minors and outlawing healthcare providers from offering CTP, with other proposed legislation (e.g., in Quebec) emerging rapidly [22]. Enforcement of provincial and municipal bans has been limited, likely owing to problems with the narrow definition of CTP used in these bans and the lack of transparency on the part of CT practitioners with regard to what their service intends to achieve [2, 23]. On October 1, 2020, the Canadian federal government tabled Bill C-6, *An Act to Amend the Criminal Code (conversion therapy)*, which is expected to be reviewed in Parliamentary committee later this year [24]. In this context, we collected data from a large national sample of sexual and gender minority men (i.e. men identifying as gay, bisexual, trans, and queer), and Two-Spirit and non-binary people, in 2019–2020, and used this survey to assess CTP and SOGIECE experiences and inform ongoing proposals for legislation and policy, including required amendments to Bill C-6. Our primary objective was to estimate the lifetime prevalence of CTP experiences among sexual minority men in Canada sampled in 2019–2020. Secondary objectives were to: a) estimate the prevalence of CTP and SOGIECE experience across sociodemographic subgroups of sexual minority men; b) describe settings, age at initiation, and duration of CTP; and c) identify health disparities among individuals who experienced CTP.

## Methods

### Participants

*Sex Now* is a serial cross-sectional survey of gay, bisexual, trans, and queer men and Two-Spirit and non-binary people in Canada conducted by the Community-Based Research Centre (CBRC). Participants are sampled from general social media platforms (Facebook, Twitter, etc.), geolocating sexual/social networking apps targeted to sexual and gender minority men (Squirt, Scruff, Grindr, etc.), CBRC's newsletter membership, other community agencies' newsletter memberships and word-of-mouth. The questionnaire is offered in Canada's two official languages, English and French, and includes a broad set of questions that correspond to community-identified health and social priorities for sexual minority men in Canada. Because there is no defined sampling frame for sexual minority men [25], and because we cannot track how many unique individuals have seen recruitment materials for *Sex Now* across the multiple sampling platforms, a response rate cannot be calculated. The *Sex Now* 2019–20 study protocol was approved by the University of Victoria Research Ethics Board. Written informed consent was documented at the start of the survey; consent from parents or guardians was not sought for participants who were minors, consistent with review/approval of the Research Ethics Board.

Data collection for the 2019–20 *Sex Now* survey occurred between November 4, 2019 and February 6, 2020. In order to be eligible to participate in the survey, participants had to: 1) self-identify as men (inclusive of trans men), non-binary (regardless of sex assigned at birth), or Two-Spirit; 2) identify as gay, bisexual, queer, or another non-heterosexual identity and/or have reported having had sex with a man (cis or trans) in the last 5 years; 3) be 15 years of age or older; 4) be living in Canada; 5) be able to provide informed consent and complete the questionnaire in either French or English, and; 6) not have already participated in the study. Written informed consent was documented at the start of the survey; consent from parents or guardians was not sought for participants who were minors. A total of N = 14,364 individuals entered the study, of which 121 did not consent to participate, and a further 1,587 people were ineligible. In total, N = 7,237 completed the entire survey (i.e., provided a response to the last question). For the present study, we subsetted the dataset to those who answered the primary outcome question regarding CTP experience (N = 9,214).

## Measures

The primary outcome for this study was derived from the question, "have you ever been exposed to any of the following conversion efforts (check all that apply)? Conversion efforts by: licensed healthcare professional (psychologist, psychiatrist, doctor), unlicensed counselor, camp, faith-based organization focused on conversion therapy, individual religious leader (i.e., not through a formal organization), and/or another religious individual." We defined conversion efforts as "attempts to change sexual orientation or gender identity [including] more organized activities (such as counseling or faith-based rituals) that are sometimes referred to as 'conversion therapy.'" Hereafter, we refer to this outcome as CTP exposure.

As a secondary outcome, we asked participants about their experiences with SOGIECE, specifically, "have you or any person with authority (parent, caregiver, counselor, community leader, etc.) ever tried to change your sexual orientation or gender identity?" Hereafter, we refer to this outcome as SOGIECE exposure.

We considered the following social-demographic variables as potentially important modifiers of exposure to CTP and SOGIECE (secondary objective a), based on previous literature [7, 8, 11, 17]: sexual identity ("how do you identify sexually?"); gender identity (non-binary, male, other); transgender status ("do you have trans experience (i.e., your gender is different than the sex you were assigned at birth)?"); age; being 'out' (told others) about one's sexual identity; race/ethnicity; Two-Spirit status (among Indigenous participants); formal educational attainment; personal annual income; area of residence (urban, suburban, rural); immigration status (immigrant, Canadian-born); and province/territory of residence.

Among those exposed to SOGIECE, we asked whether it was their sexual orientation, gender identity, or both, which were targeted by SOGIECE. Among those exposed to CTP, we asked about the following details (secondary objective b): setting/source of CTP (licensed healthcare professional, unlicensed counselor, camp, faith-based organization, individual religious leader, and/or another religious individual); age at first initiation of CTP; lifetime duration of CTP (<1 month, 1 month– 1 year, or >1 year); and lifetime number of times attending CTP (1, 2–5, 6+).

Finally, we compared psychosocial health outcomes and access to social and professional supports between those exposed to CTP or SOGIECE and those not exposed to either: feeling isolated some or all of the time ("How often do you feel isolated from others?"); left out some or all of the time ("How often do you feel left out?"); any binge alcohol use, past 6 months ("5 + drinks within 2 hours" at least once in the past 6 months); any opiate, hallucinogen, benzodiazepine, or stimulant use (hereafter "other substance use"), past 6 months (included substances: ketamine, ecstasy, crystal methamphetamines, crack, cocaine, heroin, other

prescription opioids, fentanyl, γ-Hydroxybutyric acid ("GHB"), tranquilizers or benzodiaza-pines, or psychedelics, e.g., lysergic acid diethylamide, mescaline, mushrooms); current involvement with gay, bisexual, and/or queer (GBQ) community ("What are you CUR-RENTLY involved in? (check all that apply) [Gay activism, organization, or cultural activi-ties]"); connection to GBQ community ("How connected do you feel to the following: [Gay, bi and queer men's communities?]"); and mental health service access ("In the PAST YEAR, which of the following resources have you gone to? (check all that apply) [Registered and/or peer counsellor, Indigenous elder and/or knowledge keeper, psychiatrist, psychologist]").

### Analyses

For our primary objective, we estimated the proportion of *Sex Now* participants who reported lifetime exposure to CTP, including a 95% confidence interval (CI), offered as an estimate of precision, despite not meeting the assumption of randomness in this sample (i.e., CIs should be interpreted with caution). For secondary objectives, we similarly estimated proportions with 95% CI of those exposed to CTP and SOGIECE, including by socially-defined sub-groups. Between-group comparisons were made by estimating relative risks (RR) of CTP/SOGIECE exposure, with 95% CI calculated using exact methods with the R package *Epi*. RR 95% CI that exclude 1 were interpreted as statistically significant relative differences. Duration and frequency of CTP experiences were compared across sexual orientation and gender iden-tity sub-groups. Missing, poor data quality (e.g., conflicting answers), and "prefer not to answer" responses were removed from analyses, when applicable. Because we were interested in associations between socio-demographic characteristics and CTP/SOGIECE exposure *irre-spective* of correlations between socio-demographic variables, we focused our interpretation on bivariate ("crude") analyses. Multivariable regression models were additionally used as sen-sitivity analyses, specifically to examine whether associations identified in bivariate analyses persisted after adjustment for covariates. We used Zou's method of Poisson regression with robust variance estimation, executed with the R package *geepack*, to calculate adjusted relative risks (aRR), with 95% CI [26]. All analyses were conducted in R Version 3.6.3.

## Results

### Lifetime prevalence of CTP exposure

9.9% (95% CI 9.3, 10.5) of participants sampled in *Sex Now* 2019–20 reported lifetime exposure to CTP. CTP exposure was greater among those who identified as gay (10.4%), pansexual (13.8%), or queer (15.6%), and lesser among those who identified as bisexual (8.2%) or hetero-flexible (7.7%) (**Table 1**). CTP exposure was also greater among non-binary participants (20.0%), transgender participants (19.1%), immigrants (14.6%), younger participants (e.g., 13.2% among those <20 years of age), those who are 'out' about their sexuality (10.6%), and those earning <$30,000 per year (13.6%). Lifetime prevalence of CTP exposure was lowest among white partic-ipants (8.7%); higher among South Asian (13.9%), Indigenous (13.5%), Southeast Asian (12.7%), and East Asian (11.0%) participants; and highest among Arab (22.1%), Caribbean (21.1%), Latin American (20.8%), African (18.5), and Black (18.0%) participants. CTP prevalence ranged 8.1%-14.3% across all provinces and territories, except for Quebec (6.8%) and Yukon (4.2%), though the small sample size for Yukon did not allow for precision in this estimate.

### Lifetime prevalence in SOGIECE exposure

20.6% (95% CI 19.8, 21.4) of *Sex Now* 2019–20 participants reported lifetime exposure to SOGIECE. Differences in SOGIECE exposure by sociodemographic subgroups were similar in

**Table 1. Lifetime prevalence of exposure to conversion therapy among sexual and gender minority Canadian men, by sociodemographic subgroups, 2019.**

| Variable | n (%) in SN2019[a] (*n* = 9214) | Exposure to Conversion Therapy[b], n (%) | | % Exposed to Conversion Therapy[c] (95% CI) | RR (95% CI) |
|---|---|---|---|---|---|
| | | Yes (*n* = 910) | No (*n* = 8304) | | |
| **Sexual identity[d]** | | | | | |
| Gay | 6984 (76.0) | 723 (79.7) | 6261 (75.6) | 10.4 (9.7–11.1) | 1.24 (1.06–1.45) |
| Bisexual | 2035 (22.1) | 166 (18.3) | 1869 (22.6) | 8.2 (7.0–9.5) | 0.79 (0.67–0.93) |
| Asexual | 153 (1.7) | 20 (2.2) | 133 (1.6) | 13.1 (8.4–19.7) | 1.33 (0.88–2.01) |
| Pansexual | 694 (7.5) | 96 (10.6) | 598 (7.2) | 13.8 (11.4–16.7) | 1.45 (1.19–1.77) |
| Heteroflexible | 220 (2.4) | 17 (1.9) | 203 (2.5) | 7.7 (4.7–12.3) | 0.78 (0.49–1.24) |
| Queer | 1397 (15.2) | 218 (24.0) | 1179 (14.2) | 15.6 (13.8–17.6) | 1.77 (1.53–2.03) |
| Other | 134 (1.5) | 16 (1.8) | 118 (1.4) | 11.9 (7.2–18.9) | 1.22 (0.77–1.94) |
| **Gender identity** | | | | | |
| Non-binary | 290 (3.1) | 58 (6.4) | 232 (2.8) | 20.0 (15.6–25.2) | 2.11 (1.66–2.68) |
| Other | 70 (0.8) | 12 (1.3) | 58 (0.7) | 17.1 (9.5–28.4) | 1.81 (1.08–3.04) |
| Male | 8852 (96.1) | 839 (92.3) | 8013 (96.5) | 9.5 (8.9–10.1) | Referent |
| **Transgender status** | | | | | |
| Transgender | 791 (8.7) | 151 (17.0) | 640 (7.8) | 19.1 (16.4–22.0) | 2.15 (1.83–2.52) |
| Cisgender | 8313 (91.3) | 739 (83.0) | 7574 (92.2) | 8.9 (8.3–9.5) | Referent |
| **Two-Spirit status (of Indigenous participants)** | | | | | |
| Two-Spirit | 170 (37.3) | 29 (46.8) | 141 (35.8) | 17.1 (11.9–23.7) | 1.48 (0.93–2.35) |
| Not Two-Spirit | 286 (62.7) | 33 (53.2) | 253 (64.2) | 11.5 (8.2–16.0) | Referent |
| **Age (at time of survey)** | | | | | |
| < 20 | 372 (4.0) | 49 (5.4) | 323 (3.9) | 13.2 (10.0–17.1) | 1.44 (1.03–2.03) |
| 20–29 | 2717 (29.5) | 279 (30.7) | 2438 (29.4) | 10.3 (9.2–11.5) | 1.13 (0.88–1.43) |
| 30–39 | 2362 (25.6) | 237 (26.0) | 2125 (25.6) | 10.0 (8.9–11.3) | 1.10 (0.86–1.41) |
| 40–49 | 1380 (15.0) | 138 (15.2) | 1242 (15.0) | 10.0 (8.5–11.7) | 1.10 (0.84–1.43) |
| 50–59 | 1561 (16.9) | 132 (14.5) | 1429 (17.2) | 8.5 (7.1–10.0) | 0.93 (0.71–1.21) |
| 60+ | 822 (8.9) | 75 (8.2) | 747 (9.0) | 9.1 (7.3–11.4) | Referent |
| **"Out" about sexuality** | | | | | |
| Out | 6801 (74.0) | 720 (79.5) | 6081 (73.4) | 10.6 (9.9–11.3) | 1.36 (1.16–1.59) |
| Not out | 2387 (26.0) | 186 (20.5) | 2201 (26.6) | 7.8 (6.8–9.0) | Referent |
| **Race/Ethnicity[d]** | | | | | |
| African | 81 (0.9) | 15 (1.7) | 66 (0.8) | 18.5 (11.1–29.0) | 1.89 (1.19–2.99) |
| Arab | 163 (1.8) | 36 (4.0) | 127 (1.5) | 22.1 (16.1–29.4) | 2.29 (1.70–3.07) |
| Asian | 290 (3.2) | 32 (3.5) | 258 (3.1) | 11.0 (7.8–15.4) | 1.12 (0.80–1.56) |
| Black | 172 (1.9) | 31 (3.4) | 141 (1.7) | 18.0 (12.7–24.8) | 1.85 (1.34–2.56) |
| Caribbean | 175 (1.9) | 37 (4.1) | 138 (1.7) | 21.1 (15.5–28.1) | 2.19 (1.63–2.93) |
| Indigenous | 495 (5.4) | 67 (7.4) | 428 (5.2) | 13.5 (10.7–16.9) | 1.40 (1.11–1.76) |
| Latin-American | 336 (3.7) | 70 (7.8) | 266 (3.2) | 20.8 (16.7–25.7) | 2.20 (1.77–2.74) |
| South Asian | 202 (2.2) | 28 (3.1) | 174 (2.1) | 13.9 (9.6–19.6) | 1.42 (1.00–2.01) |
| Southeast Asian | 181 (2.0) | 23 (2.5) | 158 (1.9) | 12.7 (8.4–18.7) | 1.29 (0.88–1.90) |
| White | 7687 (84.1) | 671 (74.3) | 7016 (85.2) | 8.7 (8.1–9.4) | 0.55 (0.48–0.63) |
| **Educational attainment** | | | | | |
| < College/university degree | 2053 (23.2) | 213 (25.1) | 1840 (23.0) | 10.4 (9.1–11.8) | 1.11 (0.96–1.28) |
| College/university degree | 6778 (76.8) | 635 (74.9) | 6143 (77.0) | 9.4 (8.7–10.1) | Referent |
| **Personal income, CAD** | | | | | |
| < $30,000 | 2743 (33.8) | 374 (47.3) | 2369 (32.3) | 13.6 (12.4–15.0) | 2.01 (1.70–2.37) |
| $30,000 to $59,999 | 2507 (30.9) | 221 (28.0) | 2286 (31.2) | 8.8 (7.7–10.0) | 1.30 (1.08–1.56) |

(*Continued*)

**Table 1.** (Continued)

| Variable | n (%) in SN2019[a] (*n* = 9214) | Exposure to Conversion Therapy[b], n (%) | | % Exposed to Conversion Therapy[c] (95% CI) | RR (95% CI) |
|---|---|---|---|---|---|
| | | Yes (*n* = 910) | No (*n* = 8304) | | |
| ≥ $60,000 | 2874 (35.4) | 195 (24.7) | 2679 (36.5) | 6.8 (5.9–7.8) | Referent |
| **Area of residence** | | | | | |
| Rural | 423 (4.8) | 45 (5.3) | 378 (4.7) | 10.6 (7.9–14.1) | 1.15 (0.86–1.53) |
| Suburban | 2303 (26.0) | 233 (27.5) | 2070 (25.8) | 10.1 (8.9–11.4) | 1.09 (0.95–1.26) |
| Urban | 6155 (69.3) | 569 (67.2) | 5586 (69.5) | 9.2 (8.5–10.0) | Referent |
| **Immigration status** | | | | | |
| Immigrant | 1499 (16.4) | 219 (24.3) | 1280 (15.5) | 14.6 (12.9–16.5) | 1.64 (1.43–1.89) |
| Canadian-born | 7662 (83.6) | 681 (75.7) | 6981 (84.5) | 8.9 (8.3–9.6) | Referent |
| **Province/Territory** | | | | | |
| British Columbia | 1642 (17.8) | 181 (19.9) | 1461 (17.6) | 11.0 (9.6–12.7) | Referent |
| Alberta | 1383 (15.0) | 150 (16.5) | 1233 (14.8) | 10.8 (9.3–12.6) | 0.98 (0.80–1.21) |
| Saskatchewan | 241 (2.6) | 22 (2.4) | 219 (2.6) | 9.1 (5.9–13.7) | 0.83 (0.54–1.26) |
| Manitoba | 286 (3.1) | 41 (4.5) | 245 (3.0) | 14.3 (10.6–19.1) | 1.30 (0.95–1.78) |
| Ontario | 2805 (30.4) | 302 (33.2) | 2503 (30.1) | 10.8 (9.7–12.0) | 0.98 (0.82–1.16) |
| Quebec | 2031 (22.0) | 138 (15.2) | 1893 (22.8) | 6.8 (5.8–8.0) | 0.62 (0.50–0.76) |
| Newfoundland & Labrador | 153 (1.7) | 18 (2.0) | 135 (1.6) | 11.8 (7.3–18.2) | 1.07 (0.68–1.68) |
| New Brunswick | 163 (1.8) | 15 (1.6) | 148 (1.8) | 9.2 (5.4–15.0) | 0.83 (0.51–1.38) |
| Nova Scotia | 422 (4.6) | 34 (3.7) | 388 (4.7) | 8.1 (5.7–11.1) | 0.73 (0.51–1.04) |
| Prince Edward Island | 45 (0.5) | 6 (0.7) | 39 (0.5) | 13.3 (5.5–27.5) | 1.21 (0.57–2.58) |
| Yukon | 24 (0.3) | 1 (0.1) | 23 (0.3) | 4.2 (0.2–23.1) | 0.38 (0.06–2.59) |
| Northwest Territories | 8 (0.1) | 1 (0.1) | 7 (0.1) | 12.5 (0.01–53.3) | 1.13 (0.18–7.13) |
| Nunavut | 11 (0.1) | 1 (0.1) | 10 (0.1) | 9.1 (0.5–42.9) | 0.82 (0.13–5.37) |

Note: N = 9,214; CI, confidence interval; RR, relative risk comparing prevalence of conversion therapy exposure to that of referent group; SN2019, *Sex Now* 2019.

[a] Missing, poor data quality, and "prefer not to answer" responses removed when applicable

[b] Percentage calculated using column total as denominator.

[c] Percentage calculated using row total as denominator.

[d] Responses allowed multiple-answers, overlap in responses.

direction and statistical significance as those observed for CTP, with the following exceptions (**Table 2**). Asexual participants (35.1%) and those without a college/university degree (23.4%) reported greater SOGIECE exposure than relevant comparator groups, while those living in Quebec (14.3%) and Saskatchewan (17.6%) reported lesser SOGIECE exposure than those living in British Columbia (23.7%).

## Multivariable associations between socio-demographic characteristics and CTP and SOGIECE exposure

aRRs estimating associations between CTP exposure and socio-demographic characteristics were generally similar to crude RRs in magnitude, direction, and statistical significance, except in the case of age. The aRR for those <20 years of age decreased and was no longer statistically significant (aRR = 0.71, 95% CI 0.46, 1.10 cf. crude RR = 1.44, 95% CI 1.03, 2.03), while the aRR for those 20–29 years of age decreased and became statistically significant (aRR = 0.69, 95% CI 0.52, 0.90 cf. crude RR = 1.13, 95% CI 0.88, 1.43) (**Table 3** cf. **Table 1**). aRRs estimating associations between SOGIECE exposure and socio-demographic characteristics were also

**Table 2. Lifetime prevalence of exposure to Sexual Orientation and Gender Identity Change Efforts (SOGIECE) among sexual and gender minority Canadian men, by sociodemographic subgroups, 2019.**

| Variable | n (%) in SN2019[a] (n = 9152) | Exposure to SOGIECE[b], n (%) | | % Exposed to SOGIECE[c] (95% CI) | RR (95% CI) |
|---|---|---|---|---|---|
| | | Yes (n = 1884) | No (n = 7268) | | |
| **Sexual identity[d]** | | | | | |
| Gay | 6952 (76.0) | 1449 (76.9) | 5503 (75.8) | 20.8 (19.9–21.8) | 1.05 (0.95–1.15) |
| Bisexual | 2016 (22.1) | 368 (19.5) | 1648 (22.7) | 18.3 (16.6–20.0) | 0.86 (0.77–0.95) |
| Asexual | 151 (1.7) | 53 (2.8) | 98 (1.4) | 35.1 (27.6–43.3) | 1.72 (1.38–2.15) |
| Pansexual | 688 (7.5) | 224 (11.9) | 464 (6.4) | 32.6 (29.1–36.2) | 1.66 (1.48–1.86) |
| Heteroflexible | 217 (2.4) | 29 (1.5) | 188 (2.6) | 13.4 (9.3–18.8) | 0.64 (0.46–0.90) |
| Queer | 1387 (15.2) | 498 (26.4) | 889 (12.2) | 35.9 (33.4–38.5) | 2.01 (1.85–2.19) |
| Other | 132 (1.5) | 38 (2.0) | 94 (1.3) | 28.8 (21.4–37.4) | 1.41 (1.08–1.85) |
| **Gender identity** | | | | | |
| Non-binary | 288 (3.1) | 139 (7.4) | 149 (2.1) | 48.3 (42.4–54.2) | 2.47 (2.17–2.80) |
| Other | 70 (0.8) | 26 (1.4) | 44 (0.6) | 37.1 (26.1–49.6) | 1.90 (1.40–2.58) |
| Male | 8792 (96.1) | 1719 (91.2) | 7073 (97.3) | 19.6 (18.7–20.4) | Referent |
| **Transgender status** | | | | | |
| Transgender | 778 (8.6) | 389 (21.0) | 389 (5.4) | 50.0 (46.5–53.5) | 2.83 (2.60–3.08) |
| Cisgender | 8269 (91.4) | 1460 (79.0) | 6809 (94.6) | 17.7 (16.8–18.5) | Referent |
| **Two-Spirit status (of Indigenous participants)** | | | | | |
| Two-Spirit | 169 (37.1) | 45 (36.3) | 124 (37.5) | 26.5 (20.1–33.9) | 0.96 (0.71–1.32) |
| Not Two-Spirit | 286 (62.9) | 79 (63.7) | 207 (62.5) | 27.6 (22.6–33.3) | Referent |
| **Age (at time of survey)** | | | | | |
| < 20 | 368 (4.0) | 134 (7.1) | 234 (3.2) | 36.4 (31.5–41.6) | 3.03 (2.41–3.81) |
| 20–29 | 2695 (29.4) | 691 (36.7) | 2004 (27.6) | 25.6 (24.0–27.3) | 2.13 (1.75–2.60) |
| 30–39 | 2346 (25.6) | 469 (24.9) | 1877 (25.8) | 20.0 (18.4–21.7) | 1.66 (1.36–2.04) |
| 40–49 | 1371 (15.0) | 255 (13.5) | 1116 (15.4) | 18.6 (16.6–20.8) | 1.55 (1.25–1.92) |
| 50–59 | 1557 (17.0) | 237 (12.6) | 1320 (18.2) | 15.2 (13.5–17.1) | 1.27 (1.02–1.58) |
| 60+ | 815 (8.9) | 98 (5.2) | 717 (9.9) | 12.0 (9.9–14.5) | Referent |
| **"Out" about sexuality** | | | | | |
| Out | 6775 (74.1) | 1520 (80.7) | 5255 (72.4) | 22.4 (21.5–23.5) | 1.46 (1.32–1.62) |
| Not out | 2366 (25.9) | 363 (19.3) | 2003 (27.6) | 15.3 (13.9–16.9) | Referent |
| **Race/Ethnicity[d]** | | | | | |
| African | 81 (0.9) | 28 (1.5) | 53 (0.7) | 34.6 (24.6–46.0) | 1.69 (1.25–2.29) |
| Arab | 163 (1.8) | 62 (3.3) | 101 (1.4) | 38.0 (30.7–46.0) | 1.88 (1.54–2.29) |
| Asian | 288 (3.2) | 93 (5.0) | 195 (2.7) | 32.3 (27.0–38.1) | 1.60 (1.35–1.90) |
| Black | 170 (1.9) | 56 (3.0) | 114 (1.6) | 32.9 (26.0–40.6) | 1.62 (1.30–2.01) |
| Caribbean | 171 (1.9) | 57 (3.0) | 114 (1.6) | 33.3 (26.4–41.0) | 1.64 (1.32–2.03) |
| Indigenous | 492 (5.4) | 132 (7.1) | 360 (5.0) | 26.8 (23.0–31.0) | 1.33 (1.14–1.54) |
| Latin-American | 334 (3.7) | 127 (6.8) | 207 (2.9) | 38.0 (32.8–43.5) | 1.91 (1.65–2.20) |
| South Asian | 196 (2.2) | 64 (3.4) | 132 (1.8) | 32.7 (26.2–39.8) | 1.61 (1.31–1.97) |
| Southeast Asian | 179 (2.0) | 48 (2.6) | 131 (1.8) | 26.8 (20.6–34.0) | 1.31 (1.03–1.68) |
| White | 7650 (84.3) | 1419 (75.9) | 6231 (86.4) | 18.5 (17.7–19.4) | 0.59 (0.54–0.64) |
| **Educational attainment** | | | | | |
| < College/university degree | 2046 (23.3) | 478 (26.5) | 1568 (22.4) | 23.4 (21.6–25.3) | 1.19 (1.08–1.30) |
| College/university degree | 6745 (76.7) | 1326 (73.5) | 5419 (77.6) | 19.7 (18.7–20.6) | Referent |
| **Personal income** | | | | | |
| < $30,000 | 2732 (33.8) | 738 (43.5) | 1994 (31.2) | 27.0 (25.4–28.7) | 1.66 (1.49–1.84) |
| $30,000 to $59,999 | 2499 (30.9) | 490 (28.9) | 2009 (31.4) | 19.6 (18.1–21.2) | 1.20 (1.07–1.35) |

*(Continued)*

**Table 2.** (Continued)

| Variable | n (%) in SN2019[a] (n = 9152) | Exposure to SOGIECE[b], n (%) | | % Exposed to SOGIECE[c] (95% CI) | RR (95% CI) |
|---|---|---|---|---|---|
| | | Yes (n = 1884) | No (n = 7268) | | |
| ≥ $60,000 | 2862 (35.4) | 467 (27.6) | 2395 (37.4) | 16.3 (15.0–17.7) | Referent |
| **Area of residence** | | | | | |
| Rural | 422 (4.8) | 81 (4.5) | 341 (4.9) | 19.2 (15.6–23.3) | 0.94 (0.77–1.15) |
| Suburban | 2293 (26.0) | 482 (26.6) | 1811 (25.8) | 21.0 (19.4–22.8) | 1.03 (0.94–1.13) |
| Urban | 6121 (69.3) | 4871 (69.4) | 1250 (68.9) | 20.4 (19.4–21.5) | Referent |
| **Immigration status** | | | | | |
| Immigrant | 1488 (16.3) | 431 (23.1) | 1057 (14.6) | 29.0 (26.7–31.4) | 1.53 (1.40–1.68) |
| Canadian-born | 7613 (83.7) | 1438 (76.9) | 6175 (85.4) | 18.9 (18.0–19.8) | Referent |
| **Province/Territory** | | | | | |
| British Columbia | 1630 (17.8) | 386 (20.5) | 1244 (17.1) | 23.7 (21.7–25.8) | Referent |
| Alberta | 1371 (15.0) | 304 (16.1) | 1067 (14.7) | 22.2 (20.0–24.5) | 0.94 (0.82–1.07) |
| Saskatchewan | 239 (2.6) | 42 (2.2) | 197 (2.7) | 17.6 (13.1–23.1) | 0.74 (0.56–0.99) |
| Manitoba | 285 (3.1) | 76 (4.0) | 209 (2.9) | 26.7 (21.7–32.3) | 1.13 (0.91–1.39) |
| Ontario | 2787 (30.5) | 620 (32.9) | 2167 (29.8) | 22.2 (20.7–23.8) | 0.94 (0.84–1.05) |
| Quebec | 2016 (22.0) | 288 (15.3) | 1728 (23.8) | 14.3 (12.8–15.9) | 0.60 (0.53–0.69) |
| Newfoundland & Labrador | 153 (1.7) | 33 (1.8) | 120 (1.7) | 21.6 (15.5–29.1) | 0.91 (0.67–1.25) |
| New Brunswick | 162 (1.8) | 31 (1.6) | 131 (1.8) | 19.1 (13.6–26.2) | 0.81 (0.58–1.12) |
| Nova Scotia | 422 (4.6) | 81 (4.3) | 341 (4.7) | 19.2 (15.6–23.3) | 0.81 (0.65–1.00) |
| Prince Edward Island | 44 (0.5) | 9 (0.5) | 35 (0.5) | 20.5 (10.3–35.8) | 0.86 (0.48–1.56) |
| Yukon | 24 (0.3) | 7 (0.4) | 17 (0.2) | 29.2 (13.4–51.2) | 1.23 (0.66–2.31) |
| Northwest Territories | 8 (0.1) | 3 (0.2) | 5 (0.1) | 37.5 (10.2–74.1) | 1.58 (0.64–3.89) |
| Nunavut | 11 (0.1) | 4 (0.2) | 7 (0.1) | 36.4 (12.4–68.4) | 1.54 (0.70–3.37) |

Note: N = 9,152; CI, confidence interval; RR, relative risk comparing prevalence of SOGIECE exposure to that of referent group; SN2019, *Sex Now* 2019.

[a] Missing, poor data quality, and "prefer not to answer" responses removed when applicable

[b] Percentage calculated using column total as denominator.

[c] Percentage calculated using row total as denominator.

[d] Responses allowed multiple-answers, overlap in responses.

generally similar to crude RRs in magnitude, direction, and statistical significance, except in the cases of sexual orientation, gender identity and income. The aRR for gay respondents increased and became statistically singificant (aRR = 1.21, 95% CI 1.08, 1.36 cf. crude RR = 1.05, 95% CI 0.95, 1.15). The aRR for non-binary and other gender identities both decreased and were no longer statistically significant (non-binary aRR = 1.07, 95% CI 0.91, 1.25 cf. crude RR = 2.47, 95% CI 2.17, 2.80; other aRR = 0.91, 95% CI 0.65, 1.27 cf. crude RR = 1.90, 95% CI 1.40, 2.58). Finally, the aRR for income $30,000 to $59,999 also decreased and was no longer statistically signficiant (aRR = 1.09, 95% CI 0.97, 1.23 cf. crude RR = 1.20, 95% CI 1.07, 1.35). (**Table 3** cf. **Table 2**).

## Details of CTP experiences

Among the 910 participants who were exposed to CTP, 77.3% reported that CTP exclusively targeted their sexual orientation, 5.9% reported that it exclusively targeted their gender identity, and 16.8% reported that it targeted both (**Table 4**). Most (67%) experienced CTP in a religious setting (faith-based organization, individual religious leader, or other religious individual), 30.3% with a licensed HCP, and 20.3% with an unlicensed HCP. Additionally, 72.0%

**Table 3. Multivariable-adjusted associations between sociodemographic characteristics and exposure to conversion therapy or Sexual Orientation and Gender Identity Change Efforts (SOGIECE), among sexual and gender minority Canadian men, 2019.**

| Variable | Conversion Therapy aRR (95% CI) | SOGIECE aRR (95% CI) |
|---|---|---|
| **Sexual identity** | | |
| Gay | 1.34 (1.10, 1.62) | 1.21 (1.08, 1.36) |
| Other | Referent | Referent |
| **Gender identity** | | |
| Non-binary | 1.39 (1.01, 1.92) | 1.07 (0.91, 1.26) |
| Other | 0.89 (0.47, 1.70) | 0.91 (0.65, 1.27) |
| Male | Referent | Referent |
| **Transgender status** | | |
| Transgender | 2.07 (1.64, 2.60) | 2.72 (2.41, 3.07) |
| Cisgender | Referent | Referent |
| **Age (at time of survey)** | | |
| < 20 | 0.71 (0.46, 1.10) | 2.09 (1.59, 2.74) |
| 20–29 | 0.69 (0.52, 0.90) | 1.57 (1.26, 1.96) |
| 30–39 | 0.86 (0.66, 1.13) | 1.35 (1.08, 1.69) |
| 40–49 | 1.00 (0.75, 1.33) | 1.39 (1.10, 1.76) |
| 50–59 | 0.96 (0.72, 1.27) | 1.31 (1.03, 1.66) |
| 60+ | Referent | Referent |
| **"Out" about sexuality** | | |
| Out | 1.34 (1.11, 1.63) | 1.35 (1.20, 1.51) |
| Not out | Referent | Referent |
| **Race/Ethnicity[a]** | | |
| Indigenous | 1.33 (1.03, 1.73) | 1.20 (1.02, 1.41) |
| POC | 1.64 (1.33, 2.03) | 1.64 (1.44, 1.87) |
| White | Referent | Referent |
| **Educational attainment** | | |
| < College/university degree | 0.97 (0.82, 1.14) | 1.00 (0.90, 1.11) |
| College/university degree | Referent | Referent |
| **Personal income, CAD** | | |
| < $30,000 | 1.99 (1.65, 2.39) | 1.19 (1.05, 1.34) |
| $30,000 to $59,999 | 1.36 (1.12, 1.64) | 1.09 (0.97, 1.23) |
| ≥ $60,000 | Referent | Referent |
| **Area of residence** | | |
| Rural | 1.29 (0.95, 1.75) | 1.08 (0.87, 1.33) |
| Suburban | 1.14 (0.97, 1.34) | 1.03 (0.93, 1.13) |
| Urban | Referent | Referent |
| **Immigration status** | | |
| Immigrant | 1.28 (1.13, 1.68) | 1.28 (1.13, 1.46) |
| Canadian-born | Referent | Referent |

Note: N = 9,214; CI, confidence interval; aRR, adjusted relative risk comparing prevalence of conversion therapy or sexual orientation and gender identity change efforts (SOGIECE) exposure to that of referent group (adjusted for all variables shown in table); SN2019, *Sex Now* 2019.

[a] Participants were given the option to select multiple racial/ethnic identities; owing to collinearity, identities were categorized as follows for multivariable modeling: Indigenous = any Indigenous identity, including those who are Indigenous and identified with another ethnic/racial category; People of colour (POC) = any identity other than Indigenous and white, including those who are POC and identified with the white category (i.e., multiethnic/ multiracial respondents).

**Table 4. Characteristics of conversion therapy among sexual and gender minority Canadian men, 2019.**

| Variable | n (%)[a] (N = 910) |
|---|---|
| **CTP exclusively targeting sexual orientation** | 564 (77.3) |
| **CTP exclusively targeting gender identity** | 43 (5.9) |
| **CTP targeting both sexual orientation and gender identity** | 123 (16.8) |
| **Conversion therapy via[b]** | |
| Licensed HCP | 276 (30.3) |
| Unlicensed HCP | 185 (20.3) |
| Camp | 73 (8.0) |
| Faith-based organization | 218 (24.0) |
| Individual religious leader | 334 (36.7) |
| Religious individual | 383 (42.1) |
| Other | 169 (18.6) |
| **Age at first conversion effort** | |
| < 20 | 525 (72.0) |
| 20–29 | 149 (20.4) |
| 30–39 | 40 (5.5) |
| 40–49 | 8 (1.1) |
| 50–59 | 7 (1.0) |
| 60+ | 0 (0.0) |
| **Duration of conversion effort** | |
| < 1 month | 381 (49.5) |
| 1 month– 1 year | 205 (26.7) |
| > 1 year | 183 (23.8) |
| **Conversion effort attempts** | |
| 1 time only | 269 (34.9) |
| 2–5 times | 264 (34.2) |
| > 5 times | 238 (30.9) |

Note: N = 910; CTP, conversion therapy practices; HCP, health care practitioner.

[a] Missing response removed when applicable. Percentage calculated using column total as denominator.

[b] Responses allowed multiple-answers, overlap in responses.

initiated CTP before the age of 20 years. Duration and frequency of CTP ranged widely, with 23.8% attending for longer than 1 year and 30.9% attending more than five sessions. Duration and frequency of CTP did not differ by sexual orientation; however, more transgender participants (31.2%) than cisgender particiapnts (22.2%) reported attending CTP for >1 year ($p<0.05$), and more transgender particiapnts (39.2%) than cisgender participants (28.8%) reported attending CTP >5 times ($p<0.05$).

## Psychosocial health outcomes and social and professional support access

Relative to those not exposed to CTP, those exposed reported greater experiences of isolation and feeling left out (**Table 5**). Involvement with GBQ community, connection to GBQ community, and mental health service access were also more common among those exposed to CTP as compared with those not exposed. Direction and statistical significance of comparisons in psychosocial outcomes were similar for SOGIECE exposure, with the addition of other substance use, which was more common among those exposed to SOGIECE than among those not exposed (**Table 6**).

**Table 5. Psychosocial health outcomes and social and professional support access associated with exposure to conversion therapy among sexual and gender minority Canadian men.**

| Variable | n (%) among those Exposed to CTP[a] (n = 910) | n (%) among Those Not Exposed to CTP[a] (n = 8304) | RR (95% CI) |
|---|---|---|---|
| **Psychosocial health outcomes** | | | |
| Isolated some or all of time | 495 (73.8) | 3723 (61.5) | 1.67 (1.42–1.97) |
| Left out some or all of time | 502 (74.8) | 3846 (63.4) | 1.63 (1.38–1.93) |
| Any binge alcohol use, past 6 months | 363 (53.0) | 3457 (55.8) | 0.90 (0.78–1.04) |
| Any other substance use, past 6 months[b] | 184 (26.9) | 1491 (24.1) | 1.14 (0.97–1.34) |
| **Social and professional supports** | | | |
| Involved with GBQ community | 214 (32.3) | 1132 (19.1) | 1.86 (1.60–2.17) |
| Connected to GBQ community | 247 (36.9) | 1888 (31.3) | 1.25 (1.08–1.45) |
| Accessed mental health services[c] | 380 (57.4) | 2112 (35.3) | 2.16 (1.87–2.50) |

Note: N = 9,214; CI, confidence interval; RR, relative risk comparing prevalence of conversion therapy exposure to that of referent group; CTP, conversion therapy; GBQ, gay, bi, and/or queer.

[a] Missing, poor data quality, and "prefer not to answer" responses removed when applicable. Percentage calculated using column total as denominator.

[b] Benzodiazepines, cocaine, crack, crystal, ecstasy, fentanyl, γ-hydroxybutyrate (GHB), heroin, ketamine, other opioids, psychedelics.

[c] In the past year.

**Table 6. Psychosocial health outcomes and social and professional support access associated with exposure to Sexual Orientation and Gender Identity Change Efforts (SOGIECE) among sexual minority Canadian men.**

| Variable | n (%) among those Exposed to SOGIECE[a] (n = 1884) | n (%) among Those Not Exposed to SOGIECE[a] (n = 7268) | RR (95% CI) |
|---|---|---|---|
| **Psychosocial health outcomes** | | | |
| Isolated some or all of time | 1047 (72.9) | 3155 (59.9) | 1.60 (1.44–1.78) |
| Left out some or all of time | 1062 (74.0) | 3270 (62.0) | 1.56 (1.40–1.74) |
| Any binge alcohol use, past 6 months | 841 (57.4) | 2969 (55.2) | 1.07 (0.98–1.18) |
| Any other substance use, past 6 months[b] | 403 (27.5) | 1265 (23.5) | 1.18 (1.06–1.30) |
| **Social and professional supports** | | | |
| Involved with GBQ community | 436 (31.0) | 902 (17.5) | 1.76 (1.60–1.93) |
| Connected to GBQ community | 500 (34.9) | 1630 (31.1) | 1.15 (1.04–1.26) |
| Accessed mental health services[c] | 721 (50.1) | 1759 (33.8) | 1.68 (1.54–1.84) |

Note: N = 9,152; CI, confidence interval; RR, relative risk comparing prevalence of SOGIECE exposure to that of referent group; GBQ, gay, bi, and/or queer.

[a] Missing, poor data quality, and "prefer not to answer" responses removed when applicable. Percentage calculated using column total as denominator.

[b] Benzodiazepines, cocaine, crack, crystal, ecstasy, fentanyl, γ-hydroxybutyrate (GHB), heroin, ketamine, other opioids, psychedelics.

[c] In the past year.

## Discussion

This study demonstrates that 1 in 10 sexual minority men non-randomly sampled in a Canadian community-based survey have experienced conversion therapy practices. Assuming that 4% of Canadian men are sexual minorities [27, 28], this corresponds to more than 50,000 adult men in Canada. As with previous surveys in Canada [8], we found that experiences of CTP were more prevalent among the youngest survey participants (i.e., 13% among those <20 years of age versus 9% among those 60+ years of age), despite these men having come of age during a time when visibility and social acceptance of minority sexual orientations are greater than they were for previous generations [29]. This paradoxical finding may be partially explained by a 'developmental collision', in which younger generations are coming out earlier, thereby increasing the period of potential exposure to CTP [30]. We found some support for this hypothesis in multivariable analyses, which reduced or reversed the aRR for the youngest age groups, with adjustment for a variety of covarying factors, including 'outness' (**Table 3**). Regardless of the reasons, the high prevalence of CTP among those <20 years of age suggests that existing measures to curtail CTP—such as the numerous statements issued by health professionals in opposition to CTP [1]—are insufficient.

Our CTP prevalence estimate is greater than that derived from the 2011–12 iteration of our survey (3.5%) but comparable with more recent US estimates (8–18%) [8, 17, 18]. There are several reasons to regard the 2019–20 estimate as a more valid reflection of the true prevalence of CTP among sexual minority men in Canada. First, in the 2011–12 survey, CTP was narrowly defined as "sexual repair/reorientation counseling", whereas in the 2019–20 survey, we used a broader definition of "attempts to change sexual orientation or gender identity [including] more organized activities (such as counseling or faith-based rituals) that are sometimes referred to as 'conversion therapy'", which is consistent with US survey measures and international definitions of contemporary CTP [7, 17, 18, 31]. Second, the 2019–20 survey is more recent and thus may reflect any temporal changes in CTP experiences. Third, we included and identified 9% of the 2019–20 sample as transgender and included measures of CTP targeting gender identity, allowing us to more fully capture ongoing and prevalent transgender CTP.

This is the first study (to our knowledge) to describe detailed characteristics of CTP in a large sample. Most importantly, we found that there is not a single setting where CTP occurs. Rather, these practices occur in licensed and unlicensed healthcare provider offices, camps, and organized religious settings, as well as in individual consultation with religious leaders and other religious individuals. This finding affirms the need for a multi-pronged strategy to address CTP, which may include social and therapeutic (counseling, support groups, etc.) supports for those who have experienced trauma due to CTP, legislative bans, institutional policies that create sexual and gender minority-affirming healthcare and religious environments, and communications strategies that inform youths and their families that CTP is harmful and unwarranted for those struggling with matters of sexual orientation and/or gender identity and/or expression [12].

Exposure to CTP is not evenly distributed within the population of sexual and gender minority men in Canada. Rather, experiences of CTP were more common among those with lower current levels of personal income, non-binary and transgender people, and many racialized minorities, including African, Arab, Black, Carribbean, Indigenous, Latin American, and South Asian individuals. The implications of these social inequities in CTP experience are dire and must be addressed. In the case of transgender people, new laws, policies, and interventions must be inclusive of the forms of CTP that trans people experience; unlike CTP targeting sexual orientation, most CTP targeting gender identity and expression continues to occur in licensed healthcare settings, when transgender individuals are seeking access to gate-kept

gender-affirming care [1, 4, 10, 15, 32, 33]. As currently written, federal Bill C-6 excludes forms of CTP at the hands of a practitioner who argues that they do not intend to "*change* a person's. . . gender identity" but whose goal is nevertheless to discourage or delay the adoption of gender identities not assigned at birth, as well as non-conforming gender expressions [5].

In the case of racial, ethnic, and immigrant disparities in CTP experience, we must work with racialized and immigrant communities to understand how CTP is presented in cross-cultural contexts. For example, researchers and sexual and gender minority-affirming organizations may undertake an in-depth examination of how LGBTQ2-affirming policies differentially reaches families and youth of colour [34–37]. In addition, reviews of the recent international growth of CTP point to a mechanism whereby a sizeable number sexual and gender minority immigrants may be arriving to Canada from countries where levels of sexual and gender minority stigma, and in some cases criminalization, are high, and therefore from contexts where CTP is even more common than it is in Canada [38]. Further research exploring the intersection of ethnoracial identity and immigration status in relation to CTP may help to interpret these findings. Given the overrepresentation of people of colour among those who have endured CTP, we suggest provision of CTP support services that are focused on the needs of racialized CTP survivors. We additionally acknowledge that SOGIECE was, in concept, a component of the residential schools used to strip Indigenous people of their culture; as Sarah Hunt notes, "residential schools racialized native children as 'Indians' while enforcing strict divisions between girls and boys through European dress and hairstyles, as well as physically separating them in different dorms", thereby contravening pre-colonial non-binary gender identities and roles, now referred to as Two-Spirit [14, 39]. It may be the case that one of the lasting effects of colonialism and Christianity imposed upon Indigenous peoples includes greater exposure to CTP; regardless, more work is needed to understand this critical inequity.

## Limitations

Our sample largely came from social media or dating applications and thus constitutes a non-probabilistic subset of the total target population. We have previously demonstrated that non-probability surveys tend to overrepresent employed, high-income-earning, and gay-identified sexual minorities [40]. Given that income is inversely associated with CTP exposure, in our data, this suggests that we may have underestimated the lifetime prevalence of CTP among sexual minority men in Canada. While the measure of CTP we used was more sensitive than that in the 2011–12 survey, CTP remains poorly understood, and often vaguely defined [2]. Thus, we likely undercounted some experiences of CTP, and this is an additional reason to expect that 10% is an underestimate of the true prevalence of CTP. By design, *Sex Now* does not include sexual minority women; data from the US, however, have shown comparably high rates of CTP among lesbian, bisexual, and queer women (personal communication, Christy Mallory, April 24, 2020) [18]. On this basis, we recommend that any and all proposed laws, policies, and interventions to address CTP in Canada be made gender-inclusive. While we explored a broad array of social determinants of CTP experience, we did not measure religiosity or religious affiliation/identity, which may further explain some of the variability of CTP in our sample. Finally, we used a lifetime exposure measure in this survey; therefore, we cannot know how long ago CTP occurred. The inverse relationship between age and CTP exposure suggests that contemporary CTP remains prevalent; nonetheless, additional data to account for recency of CTP are needed.

## Conclusion

Given that over 50,000 people in Canada have experienced CTP, we recommend the following actions, in collaboration with multiple levels of government, CTP survivors, and community

organizations. First, given the disproportionate burden of CTP among non-binary and trans participants in our study, legislation addressing CTP should be reviewed to ensure that it is inclusive of transgender and gender identity-based CTP. Second, given the large burden of psychosocial problems associated with CTP experience, we recommend the initiation of sexual and gender minority-affirming social and mental health supports for those who experienced trauma through CTP. The large burden of CTP we have measured in a population of sexual and gender minority men in Canada suggests that historic trauma caused by CTP will require ongoing support, over multiple generations. Finally, all people in Canada, and especially those working with youth, have a fundamental role to play in clearly and consistently communicating that lesbian, gay, bisexual, transgender, queer, and Two-Spirit experiences and identities are valued and compatible with happy and healthy lives. Only once this message is confidently and equitably expressed to youths will CTP and its counterparts in the form of SOGIECE be fully eradicated for new generations.

## Author Contributions

**Conceptualization:** Travis Salway, Olivier Ferlatte, Michael Kwag.

**Data curation:** Travis Salway, Ben Klassen, Aidan Ablona, Harlan Pruden, Jeffrey Morgan, Michael Kwag, Kiffer Card, Nathan J. Lachowsky.

**Formal analysis:** Travis Salway, Stephen Juwono, Jeffrey Morgan.

**Funding acquisition:** Travis Salway.

**Investigation:** Travis Salway, Ben Klassen, Olivier Ferlatte, Aidan Ablona, Harlan Pruden, Kiffer Card, Rod Knight, Nathan J. Lachowsky.

**Methodology:** Travis Salway, Harlan Pruden, Jeffrey Morgan, Rod Knight, Nathan J. Lachowsky.

**Project administration:** Travis Salway, Aidan Ablona, Jeffrey Morgan, Nathan J. Lachowsky.

**Resources:** Travis Salway, Michael Kwag, Nathan J. Lachowsky.

**Supervision:** Travis Salway, Rod Knight, Nathan J. Lachowsky.

**Writing – original draft:** Travis Salway.

**Writing – review & editing:** Travis Salway, Stephen Juwono, Ben Klassen, Olivier Ferlatte, Aidan Ablona, Harlan Pruden, Jeffrey Morgan, Michael Kwag, Kiffer Card, Rod Knight, Nathan J. Lachowsky.

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
