## [Decision Letter · Decision Letter 0]

24 Feb 2021

PONE-D-20-38781

Experiences with sexual orientation and gender identity conversion therapy practices among sexual minority men in Canada, 2019-2020

PLOS ONE

Dear Dr. Salway,

Thank you for submitting your manuscript to PLOS ONE. After careful consideration, we feel that it has merit but does not fully meet PLOS ONE’s publication criteria as it currently stands. Therefore, we invite you to submit a revised version of the manuscript that addresses the points raised during the review process.

We look forward to receiving your revised manuscript.

Kind regards,

Xiaozhao Yousef Yang

Academic Editor

PLOS ONE

Journal Requirements:

Reviewers' comments:

Reviewer's Responses to Questions

**Comments to the Author**

1. Is the manuscript technically sound, and do the data support the conclusions?

Reviewer #1: Partly

2. Has the statistical analysis been performed appropriately and rigorously? 

Reviewer #1: No

3. Have the authors made all data underlying the findings in their manuscript fully available?

Reviewer #1: No

4. Is the manuscript presented in an intelligible fashion and written in standard English?

Reviewer #1: Yes

5. Review Comments to the Author

Reviewer #1: Overall, the authors explored an interesting subject which has great implications to policymaking. The organization and presentation of the manuscript are very clear. Below I listed my questions and comments for the authors.

1. I would like to see more discussions about the two previous research on CTP based in Canada. Information such as how samples were collected, sample sizes, mean age of respondents, and geographic locations of the respondents if such information is available.

2. In the introduction section, the authors may also want to briefly explain why some of their focuses matter—settings of CTP occurrence, age at initiation of CTP etc. Are variations in those variables associated with variations in risks for feelings of shame, anxiety, depression, problematic substance use, suicide ideation, and/or suicide attempts—those consequences mentioned in the first paragraph? Some more discussion on this question may better highlight the importance and contribution that this study may potentially make.

3. “In total, N=7,237 provided a response to the last question”. Does it mean that 7,237 respondents answered the very last question or all questions? If it is the former, then what is the last question? Why is the very last question so important that it is worth mentioning here?

4. In the primary outcome variable, what are those response categories?

5. “Analyses were descriptive (univariate) and bivariate; multivariable analyses were not consistent with the objectives of this study, i.e., to generate statistics that inform CTP policy and intervention, rather than to identify causal effects (i.e., controlling for relationships between covariates)”. Your data are not longitudinal and therefore even if you use multivariate analysis, the results cannot reveal causality. On the other hand, why can’t multivariate analyses be used for policy making purposes? Isn’t multivariate analysis more capable of providing evidence of robust relationship than a bivariate analysis?

6. “This study demonstrates that 1 in 10 sexual minority men in Canada have experienced conversion therapy practices”. This is a very bold statement as your sample may not be nationally representative.

7. “We found that experiences of CTP were more prevalent among the youngest survey participants, despite these men having come of age during a time when visibility and social acceptance of minority sexual orientations is greater than it was for previous generations”. Could it also be that because of the greater social acceptance in recent years, the younger are more likely to come out and thus make themselves an easier target for CTP? This is why I think multivariate analyses can do a better job on informing the public and policymakers about the actual correlates of the outcome variable of interest.

8. Reading table 1 and 2, the 95% C.I. s are fine but should caution the readers that interpreting those statistics should be based on the assumption that the sample is representative of the population of interest but the samples used in the study are not. The authors may want to raise this issue in the results section.

9. As for table 3, I was wondering does “duration of conversion effort” or “conversion effort attempts” vary by sexual and gender identity?

6. PLOS authors have the option to publish the peer review history of their article (what does this mean?). If published, this will include your full peer review and any attached files.

Reviewer #1: No

---

## [Author Response · Author response to Decision Letter 0]

19 Apr 2021

April 10, 2021

Dear Dr. Yang,

On behalf of named co-authors, I am pleased to submit a revised manuscript, in which we attend to the helpful comments from you and the reviewer, as outlined below.

We look forward to hearing from you about next steps.

Sincerely,

Travis Salway, travis_salway@sfu.ca

Editor comments:

Response: Thank you. We have adjusted the formatting of the manuscript to correspond to the PLoS ONE style requirements outlined in these links. We were unable to find instructions for file naming, but have replaced each file name with the first author’s last name + “Revised manuscript marked”, “Revised manuscript clean”, and “Response to reviewers.”

Response: We have added this detail to the manuscript (p7, lines 137-138).

3. We note that you have indicated that data from this study are available upon request. PLOS only allows data to be available upon request if there are legal or ethical restrictions on sharing data publicly. For information on unacceptable data access restrictions, please see http://journals.plos.org/plosone/s/data-availability#loc-unacceptable-data-access-restrictions .

Response: We have revised the ‘Data Availability’ statement as follows: “The University of Victoria's Human Research Ethics Board has only approved storage of our data on secure university servers because the data contain potentially sensitive information about study participants. Data are available upon request through secure university servers only. Any requests to access the data can be made to University of Victoria's Human Research Ethics Board (250-472-4545 or ethics@uvic.ca). For further information about SexNow data access, including aggregated survey statistics, visit: https://www.cbrc.net/ourstats”

Reviewer comments:

Reviewer #1: Overall, the authors explored an interesting subject which has great implications to policymaking. The organization and presentation of the manuscript are very clear. Below I listed my questions and comments for the authors.

1. I would like to see more discussions about the two previous research on CTP based in Canada. Information such as how samples were collected, sample sizes, mean age of respondents, and geographic locations of the respondents if such information is available.

Response: We have added these details about SexNow 2011 and TransPulse 2019 to the Introduction (p4, lines 80-83).

2. In the introduction section, the authors may also want to briefly explain why some of their focuses matter—settings of CTP occurrence, age at initiation of CTP etc. Are variations in those variables associated with variations in risks for feelings of shame, anxiety, depression, problematic substance use, suicide ideation, and/or suicide attempts—those consequences mentioned in the first paragraph? Some more discussion on this question may better highlight the importance and contribution that this study may potentially make.

Response: Thank you for pointing out this gap in our rationale. Empirical data regarding these parameters are important when designing legislative approaches (e.g., bans) or educational campaigns to prevent CTP exposure. We have added this explanation to the corresponding section of the Introduction (pp4-5, lines 89-92).

3. “In total, N=7,237 provided a response to the last question”. Does it mean that 7,237 respondents answered the very last question or all questions? If it is the former, then what is the last question? Why is the very last question so important that it is worth mentioning here?

Response: This means that 7,237 respondents completed the entire survey, although some may have skipped questions along the way. We have revised this sentence for clarity (p7, line 140). We believe this detail is useful for readers interested in knowing the number of participants who stopped answering survey the survey between the first and last question.

4. In the primary outcome variable, what are those response categories?

Response: We have added the response categories to this part of the Methods section, for further clarity (p7, lines 145-149).

5. “Analyses were descriptive (univariate) and bivariate; multivariable analyses were not consistent with the objectives of this study, i.e., to generate statistics that inform CTP policy and intervention, rather than to identify causal effects (i.e., controlling for relationships between covariates)”. Your data are not longitudinal and therefore even if you use multivariate analysis, the results cannot reveal causality. On the other hand, why can’t multivariate analyses be used for policy making purposes? Isn’t multivariate analysis more capable of providing evidence of robust relationship than a bivariate analysis?

Response: Thank you for this suggestion. We have now analyzed multivariable models for both of the outcomes and added these results to a new table (now labeled Table 3). Corresponding methods statements have been added (p10, lines 206-210), and the multivariable/adjusted relative risks are interpreted in a new sub-section of Results (pp11-12, lines 237-254).

6. “This study demonstrates that 1 in 10 sexual minority men in Canada have experienced conversion therapy practices”. This is a very bold statement as your sample may not be nationally representative.

Response: We agree that this estimate should be interpreted with caution, given the non-probabilistic sampling methods used. We have revised this sentence (p13, lines 280-281) as follows: “This study demonstrates that 1 in 10 sexual minority men non-randomly sampled in a Canadian community-based survey have experienced conversion therapy practices.”

7. “We found that experiences of CTP were more prevalent among the youngest survey participants, despite these men having come of age during a time when visibility and social acceptance of minority sexual orientations is greater than it was for previous generations”. Could it also be that because of the greater social acceptance in recent years, the younger are more likely to come out and thus make themselves an easier target for CTP? This is why I think multivariate analyses can do a better job on informing the public and policymakers about the actual correlates of the outcome variable of interest.

Response: Thank you for this suggestion. We agree that this is a convincing explanation for the greater prevalence of CTP in the youngest age group, and we have added this suggested interpretation to the Discussion section (p13, lines 287-292), including an acknowledgment that the association between young age and CTP exposure was reduced with adjustment, notably including adjustment for outness. 

8. Reading table 1 and 2, the 95% C.I. s are fine but should caution the readers that interpreting those statistics should be based on the assumption that the sample is representative of the population of interest but the samples used in the study are not. The authors may want to raise this issue in the results section.

Response: We have added a statement acknowledging that our sample does not meet the assumption of randomness and CIs should therefore be interpreted with caution (p9, lines 194-196).

9. As for table 3, I was wondering does “duration of conversion effort” or “conversion effort attempts” vary by sexual and gender identity?

Response: Thank you for this suggestion. We have analyzed these two variables by sexual orientation and gender identity and added the results to the corresponding section of the manuscript (p12, lines 264-267).

---

## [Decision Letter · Decision Letter 1]

18 May 2021

Experiences with sexual orientation and gender identity conversion therapy practices among sexual minority men in Canada, 2019-2020

PONE-D-20-38781R1

Dear Dr. Salway,

We’re pleased to inform you that your manuscript has been judged scientifically suitable for publication and will be formally accepted for publication once it meets all outstanding technical requirements.

Kind regards,

Xiaozhao Yousef Yang, Ph.D.

Academic Editor

PLOS ONE

Additional Editor Comments (optional):

Reviewers' comments:

Reviewer's Responses to Questions

**Comments to the Author**

1. If the authors have adequately addressed your comments raised in a previous round of review and you feel that this manuscript is now acceptable for publication, you may indicate that here to bypass the “Comments to the Author” section, enter your conflict of interest statement in the “Confidential to Editor” section, and submit your "Accept" recommendation.

Reviewer #1: All comments have been addressed

2. Is the manuscript technically sound, and do the data support the conclusions?

Reviewer #1: (No Response)

3. Has the statistical analysis been performed appropriately and rigorously? 

Reviewer #1: (No Response)

4. Have the authors made all data underlying the findings in their manuscript fully available?

Reviewer #1: (No Response)

5. Is the manuscript presented in an intelligible fashion and written in standard English?

Reviewer #1: (No Response)

6. Review Comments to the Author

Reviewer #1: (No Response)

7. PLOS authors have the option to publish the peer review history of their article (what does this mean?). If published, this will include your full peer review and any attached files.

Reviewer #1: No

---

## [Editor Report · Acceptance letter]

24 May 2021

PONE-D-20-38781R1 

Experiences with sexual orientation and gender identity conversion therapy practices among sexual minority men in Canada, 2019-2020 

Dear Dr. Salway:

I'm pleased to inform you that your manuscript has been deemed suitable for publication in PLOS ONE. Congratulations! Your manuscript is now with our production department. 

Kind regards, 

on behalf of

Dr. Xiaozhao Yousef Yang 

Academic Editor

PLOS ONE